# Determination of Phloridzin and Other Phenolic Compounds in Apple Tree Leaves, Bark, and Buds Using Liquid Chromatography with Multilayered Column Technology and Evaluation of the Total Antioxidant Activity

**DOI:** 10.3390/ph15020244

**Published:** 2022-02-18

**Authors:** Anežka Adamcová, Aleš Horna, Dalibor Šatínský

**Affiliations:** 1The Department of Analytical Chemistry, Faculty of Pharmacy, Charles University, Ak. Heyrovského 1203, 500 05 Hradec Králové, Czech Republic; adamcoa1@faf.cuni.cz (A.A.); horna@radanal.cz (A.H.); 2Institute of Nutrition and Diagnostics Pardubice, Sakařova 1400, 530 03 Pardubice, Czech Republic

**Keywords:** HPLC, apple tree material, phenolic compounds, phloridzin, stationary phases, antioxidant activity

## Abstract

Apples are known to be a rich source of phenolic compounds, however detailed studies about their content in the individual parts of apple trees are reported rarely. For this purpose, we tested various stationary phases for the determination of phenolic compounds in leaves, bark, and buds. Phloridzin, phloretin, chlorogenic acid, rutin, and quercitrin were analyzed with high performance liquid chromatography coupled with diode array detection. A YMC Triart C18-ExRS 150 × 4.6 mm, 5 µm particle size analytical column with multilayered particle technology was used. The separation was performed with a mobile phase that consisted of acetonitrile and 0.1% phosphoric acid, according to the gradient program, at a flow rate of 1 mL/min for 12.50 min. The concentration of phenolic compounds from 13 cultivars was in the range of 64.89–106.01 mg/g of dry weight (DW) in leaves, 70.81–113.18 mg/g DW in bark, and 100.68–139.61 mg/g DW in buds. Phloridzin was a major compound. The total antioxidant activity was measured using flow analysis and the correlation with the total amount of phenolic compounds was found. This finding can lead to the re-use of apple tree material to isolate substances that can be utilized in the food, pharmaceutical, or cosmetics industries.

## 1. Introduction

Phenolic compounds are referred to as phytochemicals and are found in various fruits, foods, and beverages. The growing scientific interest in these compounds is focused on apples because of their beneficial effects on human health. However, there is a limited number of reports concerning the content of individual phenolic compounds in apple trees. Phenolic compounds, which are naturally occurring plant secondary metabolites, can be classified into flavonoids and non-flavonoids. Subclasses of non-flavonoids include phenolic acids, such as benzoic and cinnamic acid. Flavonoids can be divided into subgroups of flavonols, flavones, flavan-3-ols, flavanones, isoflavones, anthocyanidins, and chalcones. Phloridzin (phloretin-2′-β-D-glucopyranoside), the principal compound of our research interest, was first isolated from the bark of the apple tree by the French chemist De Koninck in 1835 [1]. It belongs to the chemical class of dihydrochalcones and consists of two aromatic rings linked by a C3 chain with a β-D-glucopyranose moiety [2]. Dihydrochalcones are biosynthesized via a phenylpropanoid pathway that is thoroughly reported in a review of Ibdah et al. [3].

Apple (*Malus domestica* Borgh.) fruits are known to be a source of phenolic compounds. Their distribution may vary in different parts of pome, pulp, and skin [4,5,6,7]. However, detailed information about the content of phenolic compounds in the individual parts of apple trees—i.e., the leaves, bark, and buds—is poor. Several reports [8,9] revealed that the plant material of apple trees is a rich source of phenolic compounds. For instance, Sowa et al. reported high values of phenolic compounds in different leaf extracts from five cultivars and compared the different extraction solvents. In ethyl acetate extracts, the content of phenolic compounds was in the range of 558.0–639.8 mg/g of dried weight (DW), while phloridzin, found as a major component, had a concentration range of 518.7–598.8 mg/g DW [8]. Another source of phloridzin includes strawberries [10], pear barks [11], and rose hips [12]. However, its values are significantly lower than in the plant material of apple trees.

Phenolic compounds have numerous benefits for human health. They can prevent coronary diseases and cancer [13], obesity and high levels of cholesterol [14], reduce inflammation [15], and act as antioxidants [8]. Phloridzin is known to have an antihyperglycemic effect. Apple leaves containing phloridzin strongly suppressed the postprandial increase in blood glucose levels in mice [16] and an antihyperglycemic effect was reported in healthy volunteers [17]. This effect is mediated by the inhibition of sodium-dependent glucose transporters (SGLTs), mainly SGLT1 in the cells of the intestinal epithelium, as is discussed in a recent review by Tian et al. [18].

To induce biological effects, phenolic compounds must be absorbed by the gastrointestinal tract. The digestion stage includes the oral, gastric, small intestine, and colonic phase. The absorption behaviors occur mainly in the small intestine. Unfortunately for phloridzin, the enzyme lactase phloridzin hydrolase (LPH) in the small intestine cleaves it (and other glycosides) to its aglycone (phloretin), reducing its bioavailability [5]. However, the development of acylated phloridzin derivatives using long chain fatty acids may solve this issue [19]. More investigation is still needed.

Reverse-phased high performance liquid chromatography (RP-HPLC) belongs to the commonly used analytical methods for the analysis of phenolic compounds in natural products. Octadecyl-silica (C-18) columns with fully porous 5 μm particles were the most frequently used stationary phases for the separation of phenolic compounds in recent years. Longer columns (250 mm) are suitable to separate a higher number of phenolic compounds from raw plant material. The gradient elution is often needed to separate phenolic compounds in the form of glycosides (coupled with a sugar moiety) and aglycons (in the absence of sugar moiety) because of their different lipophilic properties. The mobile phase consists of an organic part (acetonitrile, methanol) and an acidified water part. Detection is usually carried out by a diode array detector (DAD) [6,8,9], mass spectrometry (MS) [10], or electrochemical detection (ECD) [20]. A short list of published HPLC conditions to separate phenolic compounds in the plant material of apple trees is summarized in Appendix A. The phenolic compounds analyzed in our study are depicted in Figure 1.

As a novelty, this study is intended to report new information about the phenolic compounds profile in non-fruit material using a fast chromatography method that employs a new type of multilayered organic–inorganic hybrid silica particle stationary phase YMC Triart C18. This reversed stationary phase was compared with others. The unique technology is based on multilayered particles that are produced via a tightly controlled granulation technology that has been adapted from microreactor technology. The mentioned stationary phase can be used across a broad pH range of 1–12 and provides reduced flow resistance and high separation performance in the chromatography system, probably due to high carbon loading (25%) [21]. A study using this type of stationary phase to separate phenolic compounds from plant material has been reported [6].

The total phenolic content can be measured spectrophotometrically [8]. The disadvantage of this method is the interference of substances that can reduce chromogenic compounds, which leads to the overestimation of the antioxidant capacity and affects the results [22,23]. In contrast, electrochemistry detection provides new possibilities for measuring the total phenolic content because of its selectivity. A fast screening method to obtain the total antioxidant/phenolic compounds with the use of electrochemistry can be called an electrochemical index (EI) [24].

The aim of this study was to analyze different extracts from leaves, bark, and buds from 13 apple tree cultivars grown in the Czech Republic. The main part was focused on the optimization and validation of the chromatography method with various stationary phases to separate the phenolic compounds in the methanol extracts of plant material. The next objective was to evaluate the phenolic profile in the plant material of the tested cultivars and to evaluate the total antioxidant activity.

## 2. Results and Discussion

### 2.1. Optimization of the Chromatographic Separation Conditions

In this study, nine various stationary phases with different polarities and different polar-embedded functional groups were tested to obtain fast separation with a high resolution of the peaks and sufficient peak symmetry: YMC Triart C18 ExRS (150 × 4.6 mm, 5 µm particle size), YMC Triart PFP plus (150 × 4.6 mm, 5 µm particle size), YMC Triart C18 (150 × 4.6 mm, 5 µm particle size), Discovery^®^ HS C18 (150 × 4.6 mm, 5 µm particle size), Luna Omega Polar C18 (150 × 4.6 mm, 5 µm particle size), and Ascentis Express RP amide (150 × 4.6 mm, 2.7 µm particle size) analytical columns were used. Three stationary phases with technology of core-shell particles: Kinetex^®^ F5 100A (150 × 4.6 mm, 2.6 µm particle size); Kinetex^®^ C18 100A (150 × 4.6 mm, 2.6 µm particle size); and Kinetex^®^ 100A Biphenyl (150 × 4.6 mm, 5 µm particle size) were tested to improve the separation efficiency and to shorten the analysis time. The separation was done using a mobile phase that consisted of an acetonitrile (A) and water solution of 0.1% phosphoric acid (B, pH = 2.2) with a flow rate of 1 mL/min, a temperature in the column space of 30 °C, and a gradient mode reported in Section 3.5. A universal detection wavelength 254 nm was used for the visualization of all peaks at one chromatogram. During the short optimization, the concentration of acetonitrile at the beginning of the gradient elution was tested in the range of 2–20%. The low content of acetonitrile prolonged the whole analysis time while the higher content caused co-elution of the chlorogenic acid with a peak at the void volume on some stationary phases. Therefore, 10% of the acetonitrile was found to be optimal for the screening of the tested stationary phases. The column oven temperature was set to 30 °C from the tested range of 30–60 °C. A higher temperature of 60 °C accelerated the last peak of phloretin from 10.45 min to 10.20 min. Thus, the shortening effect on the whole analysis time was negligible. Therefore, a higher temperature was not used because of the thermostability of the silica-based columns. The chromatograms of the mixed standard solution, using all the tested columns, are presented in Appendix A. Chromatogram of standard solution using YMC Triart C18 ExRS (150 × 4.6 mm, 5 µm particle size) analytical column is presented in Figure 2.

Within the optimization of the method, peak symmetry (A_R_), resolution (R_s_), retention times (t_R_) of all analytes, and peak capacity (P_c_) were critically evaluated. To be more specific, columns YMC Triart C18 (100 × 4.6 mm, 5 µm particle size) and YMC Triart PFP plus (150 × 4.6 mm, 5 µm particle size) showed low resolution for the phloridzin and quercitrin peaks (R_S_ = 2.90 and 1.68). Discovery^®^ HS C18 (150 × 4.6 mm, 5 µm particle size) showed bad symmetry for the phloridzin (A_R_ = 1.65) and phloretin (A_R_ = 1.76) peaks. However, very good resolution, selectivity, and symmetry of the peaks were observed in the core-shell columns, Kinetex 100A Biphenyl (150 × 4.6 mm, 5 µm particle size) and Kinetex^®^ C18 100A (150 × 4.6 mm, 2.6 µm particle size). These columns, together with Ascentis Express RP amide (150 × 4.6 mm, 2.7 µm particle size), generated a higher backpressure, which was close to the maximum working range of the LC instrument. Kinetex^®^ F5 100A (150 × 4.6 mm, 2.6 µm particle size) provided a worse resolution for chlorogenic acid with an unknown peak (R_s_ = 1.35). YMC Triart C18 ExRS (150 × 4.6 mm, 5 µm particle size) and Luna Omega Polar C18 (150 × 4.6 mm, 5 µm particle size) had similar performances, however, a longer analysis time was observed in the second column mentioned. The tested columns separated the phenolic compounds in the following order: (1) chlorogenic acid, (2) rutin, (3) quercitrin, (4) phloridzin, and (5) phloretin, except for the column Ascentis Express RP amide (150 × 4.6 mm, 2.7 µm particle size), in which the order of the phenolic compounds was as follows: (1), (2), (4), (3), and (5). Finally, the YMC Triart C18 ExRS (150 × 4.6 mm, 5 µm particle size) analytical column was used due to its short analysis time, sufficient resolution, peak symmetry, and low flow resistance for method validation and subsequent analysis of plant extracts. The separation conditions of all the tested columns are thoroughly summarized in Appendix A. The separation of standard solution under the optimized and validated conditions on the YMC Triart C18 ExRS analytical column is depicted in Figure 2.

### 2.2. Optimization of Extraction Procedure

The high extraction yield was the crucial factor in obtaining the maximum phenolic compounds from the raw plant material during the preparation of the sample. First, the extraction of 0.1 g of dry plant and roughly homogenized material was carried out using 1 mL of methanol with 0.1% formic acid (99.9:0.1, (*v/v*)). The extraction recovery of this procedure showed a high yield but worse repeatability due to an incorrect homogenization using a NutriBullet mixer. Therefore, the grinding mortar was used to thoroughly homogenize the dry plant material into fine power. Using this fine powder improved the repeatability of the whole extraction procedure. In the next optimization step, the volume of the extraction solvent was increased from 1 to 2 mL and, simultaneously, the amount of dry material was decreased from 0.1 g to 0.05 g. The reduction of the extracted plant mass and the increase of the extraction solvent resulted in a higher extraction yield and recovery that showed satisfactory results for subsequent method validations. Further increasing the extraction solvent volume resulted in more dilution of the minor phenolic compounds and thus decreased the signal for chlorogenic acid and phloretin. Therefore, the volume of 2 mL of solvent was finally used. In the next step, the influence of the pH conditions on the extraction yield of all the phenolic compounds were tested by comparison of 0.1% formic acid (pH = 2.75) and 0.1% acetic acid (pH = 3.25). The type of acid used for extraction was evaluated for nine different apple leaf cultivars. Based on the results (Appendix A), the methanol with 0.1% formic acid was found to be more efficient in terms of the extraction yield and, therefore, was finally used as a solvent for all the plant materials.

### 2.3. Validation of the Method

The following validation parameters were evaluated prior to the analysis of the plant material: linearity, repeatability, recovery, method precision, and limit of detection (LOD) and quantification (LOQ).

Linearity was measured at seven concentration levels between 2–250 mg/L (phloretin, rutin, chlorogenic acid, quercitrin) and five concentration levels between 1000–8000 mg/L (phloridzin). The higher concentration range of phloridzin was set according to its expected higher occurrence in apple tree material. All the compounds showed a good correlation coefficient of the calibration curve.

The method repeatability was performed using six injections of the mixed standard solution at three concentration levels. The relative standard deviation (RSD in %) ranged from 0.46 to 1.75% (phloretin, rutin, chlorogenic acid, and quercitrin) and from 0.34% to 0.76% (phloridzin).

The extraction recovery of five phenolic compounds was investigated. The results showed that recovery ranged from 88.78% (phloridzin) to 123.21% (chlorogenic acid). The RSD values of precision varied from 2.07% (phloridzin) to 4.56% (phloretin).

All the validation results fulfilled the criteria for quantification of the phenolic compounds in real plant extracts. The calibration range was sufficient to determine the wide range of different concentrations and the limit of quantification was sufficiently sensitive to calculate really low concentrations. These results show that the developed HPLC-DAD method is suitable for performing routine analyses of plant extracts. All details and validation parameters are reported in Table 1.

### 2.4. Determination of the Phenolic Profile in Apple Leaves, Bark, and Buds

The developed and validated HPLC method was applied for the determination of phenolic compounds in the methanol extracts of apple leaves, bark, and buds. The profiles of the phenolic compounds in cvs. ‘Bohemia Gold’, ‘Fragrance’, ‘Gloster’, ‘Goldstar’, ‘James Grieve’, ‘Melodie’, ‘Melrose’, ‘Meteor’, ‘Průsvitné letní’, ‘Red Bilt’, ‘Rubinola’, ‘Spartan’, and ‘Topaz’ were evaluated. To show the analysis of the plant extract, a chromatogram of apple bark extract at different wavelengths according to the absorption maxima of the phenolic compounds is depicted in Figure 3.

The sum of all the phenolic compounds in apple leaves, bark, and buds is summarized in Table 2. The reported results indicate that the plant material of an apple tree is an important source of phenolic compounds. Our results are in good agreement with results reported elsewhere [8,9,25].

The total content of the phenolic compounds in apple leaves is summarized in Table 2—in the first column—and in more detail with respect to the particular phenolic compounds in Table 3. The concentration range of all the determined phenolic compounds ranged from 54.68 mg/g DW (cv. ‘Goldstar’) to 106.81 mg/g DW (cv. ‘Topaz’). For comparison, Sowa et al. found a total amount of phenolic compound (p-hydroxybenzoic acid, chlorogenic acid, hyperoside, isoquercitrin, quercitrin, and phloridzin) in the range of 257.8 mg/g DW–320.4 mg/g DW in apple leaf methanolic extracts [8]. Táborský et al. observed the phenolic composition (phloridzin, phloretin, chlorogenic acid, and rutin) of individual parts of apple trees during the vegetation period. In apple leaf methanolic extracts, the highest values were found in September in cv. ‘Opal’ at a concentration of 94.58 mg/g DW [25]. The major phenolic compound of all the cultivars was definitely phloridzin at a concentration range of 46.43 mg/g DW (cv. ‘Goldstar’)–98.51 mg/g DW (cv. ‘Spartan’), followed by quercitrin 1.89 mg/g DW (cv. ‘Meteor’)–11.29 mg/g DW (cv. ‘Rubinola’), rutin 0.70 mg/g DW (cv. ‘James Grieve’)–3.83 mg/g DW (cv. ‘Rubinola’), chlorogenic acid 0.32 mg/g DW (cv. ‘Rubinola’)–1.18 mg/g DW (cv. ‘Topaz’), and phloretin 0.28 mg/g DW (cv. ‘Spartan’)–0.54 mg/g DW (cv. ‘Gloster’). The graphical evaluation of the concentrations of individual phenolic compounds is shown in Appendix A.

The results of the content of all the determined phenolic compounds in apple bark are shown in Table 2—in the second column—and are further detailed in Table 4. The concentration range of all the phenolic compounds ranged from 70.81 mg/g DW (cv. ‘Meteor’) to 113.18 mg/g DW (cv. ‘Spartan’). Táborský et al. determined the highest values of phenolic compounds in apple bark methanolic extracts at a concentration of 95.85 mg/g DW [25]. Also in bark, phloridzin represented the major component of all the cultivars at a concentration range of 54.52 mg/g DW (cv. ‘Goldstar’)–102.69 mg/g DW (cv. ‘Spartan’). Compared to leaves, a significant increase of rutin was observed. Its concentration range was from 10.68 mg/g DW (cv. ‘Gloster’) to 26.03 mg/g DW (cv. ‘Goldstar’). The concentration of quercitrin ranged from 0.73 mg/g DW (cv. ‘Meteor’) to 3.06 mg/g DW (cv. ‘Topaz’). On the contrary, phloretin was not detected in any cultivar, which is in agreement with other studies [25]. Chlorogenic acid was found only in one cultivar; however, its concentration was low (0.22 mg/g DW in cv. ‘Goldstar’). The graphical evaluation of the concentrations of individual phenolic compounds is shown in Appendix A.

The content of all the phenolic compounds in apple buds is shown in Table 2—in the third column—and are further detailed in Table 5. Apple buds represent the highest source of phenolic compounds, especially phloridzin, in this study. Only ten cultivars of apple buds were collected due to the lack of others. The values of the phenolic compounds ranged from 100.67 mg/g DW (cv. ‘James Grieve’) to 139.61 mg/g DW (cv. ‘Rubinola’). Phloridzin was found at a concentration range of 88.29 mg/g DW (cv. ‘James Grieve’)–113.80 mg/g DW (cv. ‘Bohemia Gold’). The concentration range of quercitrin was from 7.00 mg/g DW (cv. ‘Melrose’) to 20.45 mg/g DW (cv. ‘Rubinola’), chlorogenic acid was from 1.54 mg/g DW (cv. ‘James Grieve’)–6.68 mg/g DW (cv. ‘Průsvitné letní), and rutin was from 0.93 mg/g DW (cv. ‘Melrose’)–3.22 mg/g DW (cv. ’Melodie’). Phloretin was not found in any cultivar, which is consistent with other studies [25]. The graphical evaluation of the concentrations of individual phenolic compounds is shown in Appendix A.

To compare our results of leaves, bark, and buds with the content of phenolic compounds in apple fruit, we used the study by Liaudanskas et al., who found the content of phenolic compounds (procyanidins, chlorogenic acid, phloridzin) at the concentration range of 1713.2 μg/g DW–4137.7 μg/g DW—and phloridzin covered the concentration range of 75.4 μg/g DW–151.7 μg/g DW and was not the main phenolic compound [6]. Lata et al. found phloridzin at the concentration range of 710 μg/g DW–242 μg/g DW in peels and 110 μg/g DW–430 μg/g DW in the whole apple fruit [4]. Similar data on apple fruits is presented in a study of Táborský et al. [25]. All the studies confirmed there were several times higher levels of phloridzin in plant material than in fruits.

The physiological role of phenolic compounds in plants is in defense mechanisms [3]. This may be an explanation for why leaves, bark, and buds are richer in phenolic compounds than fruit. The tested material is exposed to long-term stress (for example intensive sunlight, pests), which can lead to a greater accumulation of phenolic compounds to protect the plant.

### 2.5. Determination of Total Antioxidant Activity

The total amount of antioxidants in cvs. ‘Bohemia Gold’, ‘Fragrance’, ‘Gloster’, ‘Goldstar’, ‘James Grieve’, ‘Melodie’, ‘Melrose’, ‘Meteor’, ‘Průsvitné letní’, ‘Red Bilt’, ‘Rubinola’, ‘Spartan’, and ‘Topaz’ was measured using the flow analysis method coupled with a coulochem electrochemical detector. For this purpose, the FIA-CoulArray to determine the total antioxidant activity was used for each extract. The example of FIAgram is shown in Appendix A. This method offers to compare cultivars and their total antioxidant activity in quick time. The 70 s analysis time was performed because no separation was necessary. On the other hand, the absence of a column means the impossibility of determining the antioxidant activity of individual phenolic compounds. The total antioxidant activity of all the studied material (leaves, bark, buds) expressed in μC is shown in Appendix A. In the leaf extracts, the highest values were measured in cv. ‘Gloster’ and the lowest in cv. ‘Goldstar’. The correlation between the sum of phenolic compounds measured via HPLC-DAD and the total antioxidant activity via FIA-CoulArray is summarized in Figure 4. This finding can lead to the replacement of the HPLC method, especially in the case of the rapid selection of extracts with high amounts of antioxidants. Advantageously, it can be used in laboratories with high sample throughput. A significant correlation was found for leaves and bark, while the buds’ antioxidant activity showed a poor correlation with the sum of phenolic compounds. We speculate that the reason for the poor correlation of the total antioxidant activity in buds may be caused by other unknown interfering substances that negatively affected the electrochemical detection.

### 2.6. Statistical Evaluation

There was found to be a significant difference at level 0.05 in phloridzin concentrations among leaves, bark, and buds using the ANOVA test. By the Tukey method it was found that the concentration of phloridzin in buds is higher and different from leaves and bark, which have comparable concentrations.

## 3. Materials and Methods

### 3.1. Chemical and Reagents

All standards, reagents, and solvents were of analytical grade. The standards were: phloridzin dihydrate 99%, phloretin ≥99%, chlorogenic acid ≥95%, quercitrin hydrate ≥78%, and rutin hydrate ≥94%, which were purchased from Sigma-Aldrich Chemie GmbH, Darmstadt, Germany. All other chemicals and reagents (acetonitrile, methanol, phosphoric acid, formic acid) were also obtained from Sigma-Aldrich (St. Louis, MO, USA, Merck KGaA, Darmstadt, Germany). The ultrapure water was purified through a Milli-Q system (Millipore, Bedford, MA, USA).

### 3.2. Raw Plant Material of Apple Trees

The plant material, leaves, bark, and buds of thirteen apple tree cultivars of ‘Bohemia Gold’, ‘Fragrance’, ‘Gloster’, ‘Goldstar’, ‘James Grieve’, ‘Melodie’, ‘Melrose’, ‘Meteor’, ‘Průsvitné letní’, ‘Red Bilt’, ‘Rubinola’, ‘Spartan’, and ‘Topaz’ were collected in Hradec Králové, Roudnička, Czech Republic in July 2018 (leaves), September 2018 (bark), and February 2019 (buds). Raw material was dried at a room temperature of 22 °C until extraction. Authentication was confirmed by experts from Research and Breeding Institute of Pomology Holovousy Ltd. (Holovousy, Czech Republic).

### 3.3. Preparation of the Standard Solutions for Method Optimization

A standard stock solution of 0.5 mg/mL was individually prepared for each reference standard of the phloridzin, phloretin, chlorogenic acid, rutin, and quercitrin in a 1 mL of methanol (100 *v*/*v*). All the stock solutions were stored at 4 °C in a refrigerator until use. The mixed standard solution was prepared by mixing 50 μL of phloridzin and 25 μL of the remaining standard solutions. The concentration of the phloridzin standard was 500 mg/L and the remaining standards were 25 mg/L in the mixture. This mixed standard solution was used for the HPLC method development and optimization of the separation conditions.

### 3.4. Extraction of Phenolic Compounds

The methanol extracts were prepared by weighting 0.05 g of dried powdered leaves, bark, and buds in a 2 mL centrifugation tube and a solution of 2 mL of methanol and formic acid 0.1% (*v/v*) was added. The tube was placed into an ultrasonic bath (30 min) (Sonorex Super RK 100, Bandelin, Berlin, Germany), followed by centrifugation at 5000 rpm (2320× *g*) for 15 min (MPW-260, MPW MED. Instruments, Warszawa, Poland). The extracts were filtered through 0.22 µm polytetrafluoroethylene (PTFE) membrane filters in 1.5 mL vials and stored in a refrigerator at 4 °C until analysis.

### 3.5. HPLC Equipment and Analysis

The analysis of the phenolic compounds was performed with a Shimadzu LC-10 AD (Shimadzu Corporation, Kyoto, Japan) HPLC system equipped with an LC-10AD binary solvent delivery module, an SIL-HTA autosampler, a DGU-14A online degasser, an SPD-M10A DAD detector, and a CTO 10AC column oven. The data acquisition and data evaluation were performed with the Shimadzu “LC Lab-Solution” software (Shimadzu Corporation, Kyoto, Japan).

The chromatography separations were tested using the following columns: Ascentis Express RP amide (150 × 4.6 mm, 2.7 µm particle size), Discovery^®^ HS C18 (150 × 4.6 mm, 5 µm particle size), Kinetex^®^ F5 100A (150 × 4.6 mm, 2.6 µm particle size), Kinetex^®^ C18 100A (150 × 4.6 mm, 2.6 µm particle size), Kinetex^®^ 100A Biphenyl (150 × 4.6 mm, 5 µm particle size), Luna Omega Polar C18 (150 × 4.6 mm, 5 µm particle size), YMC Triart C18 ExRS (150 × 4.6 mm, 5 µm particle size), YMC Triart PFP plus (150 × 4.6 mm, 5 µm particle size), and YMC Triart C18 (150 × 4.6 mm, 5 µm particle size).

The mobile phase consisted of an acetonitrile (A) and water solution of 0.1% phosphoric acid (B, pH = 2.2) at a flow rate of 1 mL/min according to the following elution gradient program: 0.01–10 min 10% mobile phase B, 10–10.2 min 50% mobile phase B, and 10.2–12.5 min 90% mobile phase B. The detection was performed with DAD at wavelengths of 280 nm, 327 nm, and 354 nm. The injection volume was 1 μL and the temperature of the column space was 30 °C.

The quantitative evaluation of the peaks was calculated using the method of external standard:$$c_{x}=\frac{A_{x}}{A_{ST}}*\;c_{ST}$$

The concentration of a compound was converted to the amount of extraction solvent used, followed by the conversion to the real sample weight with the correction to the purity of the used standard (Section 3.1). The total amount of individual phenolic compounds was expressed in mg/g of dried weight of plant material.

The chromatograms were evaluated at three different detection wavelengths because of the different spectral characteristics of the analyzed compounds. The wavelength of 280 nm was used for the determination of phloridzin and phloretin, 327 nm was used for chlorogenic acid, and 354 nm for the determination of rutin and quercitrin. The identification of the compounds in the plant extracts was achieved by comparing their retention time and absorbance spectrum with those of the standard solutions. The peak purity was checked to control the co-elution with other potential interferences from plant material.

### 3.6. Validation of the Method

The analytical procedure was validated according to International Conference on Harmonization of Technical Requirements for Registration of Pharmaceuticals for Human Use (ICH) guidelines [26].

For the determination of the linear range, a mixture of phloretin, rutin, chlorogenic acid, and quercitrin was prepared at a concentration of 2, 20, 50, 100, 150, 200, 250 mg/L, and phloridzin at a concentration of 1000, 2000, 4000, 6000, 8000 mg/L. The repeatability was performed using the mixed standard solution at levels of 20, 100, and 250 mg/L (phloretin, rutin, chlorogenic acid, and quercitrin), and 1000, 4000, and 8000 mg/L (phloridzin).

The recovery of the extraction procedure was carried out on eight sample solutions. Six real samples were prepared with the addition of the known concentration of standard solution as well as two samples without the addition. All these samples were extracted by the same procedure as described in Section 3.4. Each sample was measured in three repetitions. The recovery R (%) was calculated as the ratio of the spiked and un-spiked samples of extract to the peak area of the standard.

The precision of the extraction procedure was performed with eight sample solutions prepared as described in Section 3.4. Each sample was injected three times.

LOD and LOQ was calculated as the minimum concentration, yielding a signal-to-noise ratio standard deviation equal to three (3σ) and ten (10σ), respectively.

### 3.7. The Total Antioxidant Activity

The total antioxidant activity of the plant extracts was carried out with the system consisting of two solvent delivery pumps (Model 582 ESA Inc., Chelmsford, MA, USA) with an autosampler (Model 542 HPLC, ESA Inc., Chelmsford, MA, USA) and a CoulArray electrochemical detector (Model 5600A, ESA Inc, Chelmsford, MA, USA). The determination was performed via flow injection analysis (FIA) in the absence of the chromatographic column in the system. The detector includes one flow cell (Model 6210 ESA, Chelmsford, MA, USA) that consists of four working porous graphite electrodes, each with two auxiliary and two dry Pd/H_2_ reference electrodes. The potentials of 200, 400, 600, and 800 mV were applied. The data acquisition and data evaluation were performed with the CoulArray^®^ Data Station.

The determination of the total antioxidant activity was performed with a carrier solution that consisted of 0.05 mol/L KH_2_PO_4_ in water and 10% ACN (pH = 4.8) at a flow rate of 1 mL/min and a temperature of 30 °C. The analysis time was 70 s. The total antioxidant activity in 20-times diluted extracts was recorded as the mean of total microcoulombs (μC) and was calculated by the automatic integration of the resulting peak areas.

## 4. Conclusions

In the present paper, we tested nine various stationary phases for the chromatographic separation of phenolic compounds in reversed phase mode. All tested stationary phases were able to separate the phenolic compounds sufficiently in a relatively short time. One stationary phase with an embedded amide polar group, Ascentis Express RP-amide, showed a change in the elution order of phloridzin and quercitrin. Due to the short analysis time, sufficient resolution, peak symmetry, and low flow resistance, a new type of multilayered organic–inorganic hybrid silica particle stationary phase YMC Triart C18 ExRS (150 × 4.6 mm, 5 µm particle size) was used. All the tested YMC Triart columns did not exceed the flow resistance of 10 MPa in the HPLC system. The validated HPLC-DAD method was applied to study the profile of the phenolic compounds (phloridzin, phloretin, chlorogenic acid, rutin, and quercitrin) in the non-fruit material of apple trees.

A solid–liquid extraction using an ultrasonic bath and a solution of methanol and formic acid followed by filtration was found to be optimal for efficient extraction. Especially given that thorough homogenization must be performed to get a higher extraction yield and satisfactory recovery results.

There is an increasing demand for natural products with a high content of healthy substances. Our results open new insights into the distribution of the phenolic composition in apple leaves, bark, and buds. The several times higher values of phenolic compounds in these materials in comparison with fruit (mg vs. µg) can be attractive for new nutraceutical product development. The isolation of bioactive compounds from abundant and easily available leaves into pharmaceuticals or nutraceuticals provides great potential to improve human health, prevent infections, or contribute to recovery from illness. Especially given that phloridzin, the major compound in the studied material, can be used to correct hyperglycemia.

As a novelty, this research also reported other possibilities for determining the total amount of antioxidants in extracts by flow injection analysis coupled with the CoulArray detector. The method would allow companies and laboratories with high sample throughput to quickly select extracts rich in antioxidant compounds. It can have a practical impact on the fast evaluation of plant extracts in the absence of time-consuming chromatography separation. Unfortunately, the number of FIA–CoulArray publications is limited and should be further investigated.

## Figures and Tables

**Figure 1 pharmaceuticals-15-00244-f001:**
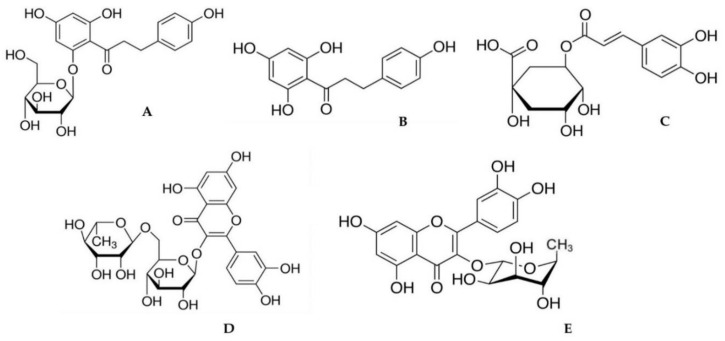
Chemical structure of (**A**) phloridzin, (**B**) phloretin, (**C**) chlorogenic acid, (**D**) rutin, and (**E**) quercitrin.

**Figure 2 pharmaceuticals-15-00244-f002:**
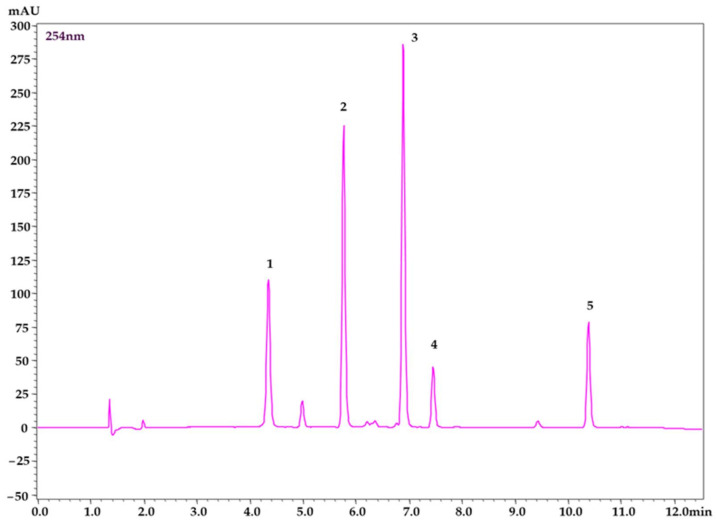
Chromatogram of mixed standard solution using YMC Triart C18 ExRS (150 × 4.6 mm, 5 µm particle size) at a universal wavelength of 254 nm. (1) chlorogenic acid, (2) rutin, (3) quercitrin, (4) phloridzin, (5) phloretin.

**Figure 3 pharmaceuticals-15-00244-f003:**
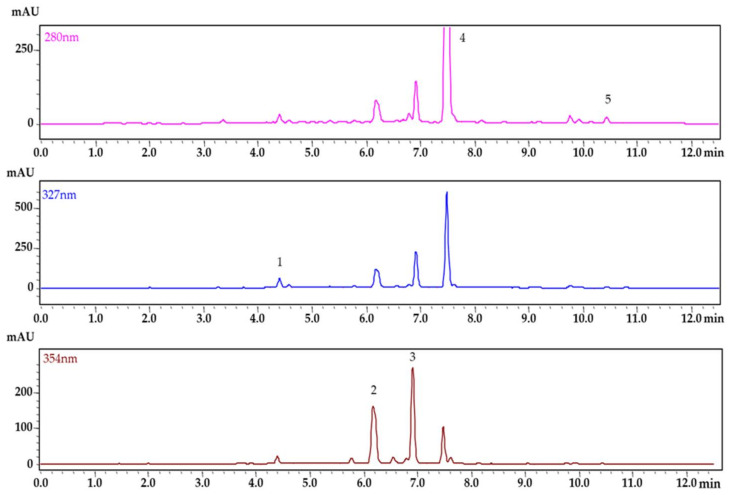
Chromatogram of phenolic compounds from apple bark extract. (1) chlorogenic acid, (2) rutin, (3) quercitrin, (4) phloridzin, (5) phloretin.

**Figure 4 pharmaceuticals-15-00244-f004:**
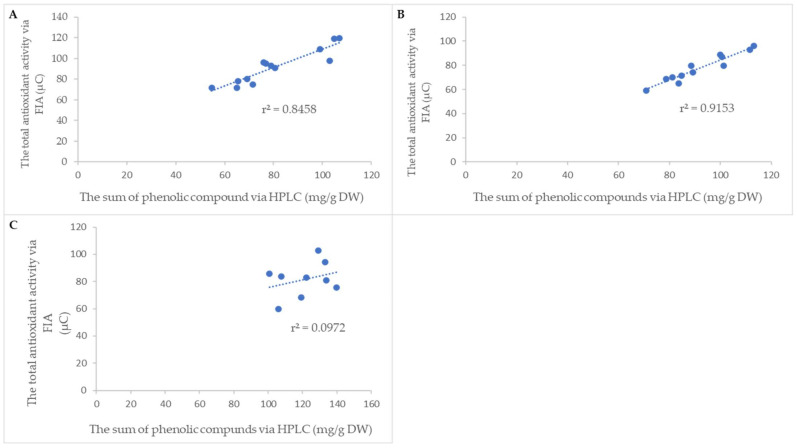
Relationship between the total antioxidant activity via FIA and the sum of phenolic compounds via HPLC in (**A**) leaves, (**B**) bark, and (**C**) buds.

**Table 1 pharmaceuticals-15-00244-t001:** Validation parameters of individual analytes using YMC Triart C18 ExRS (150 × 4.6 mm, particle size 5 μm) column.

Analyte	t_R_ ^a^ (Min)	(r^2^) ^b^	Calibration Range (mg/L)	Recovery (%)	Precision (RSD, %)	Repeatability (RSD, %)	LOD ^e^ (mg/L)	LOQ ^f^ (mg/L)
Chlorogenic acid	4.48	0.997	2–250	123.74	2.40	1.37; 0.72; 1.59 ^c^	0.078	0.260
Rutin	6.29	0.998	2–250	97.25	4.21	0.46; 0.91; 1.67 ^c^	0.145	0.434
Quercitrin	6.90	0.996	2–250	86.54	3.23	1.16; 0.90; 1.75 ^c^	0.146	0.482
Phloridzin	7.55	0.994	1000–8000	88.74	2.07	0.34; 0.51; 0.76 ^d^	0.080	0.263
Phloretin	10.52	0.998	2–250	90.44	4.56	0.47; 0.68; 1.62 ^c^	0.098	0.324

^a^ Retention time, ^b^ Correlation coefficient, ^c^ Concentration level of 20, 100, 250 mg/L, ^d^ Concentration level of 1000, 4000, 8000 mg/L, ^e^ Limit of detection, ^f^ Limit of quantification.

**Table 2 pharmaceuticals-15-00244-t002:** Content of all phenolic compounds expressed in mg/g DW obtained from apple leaves, bark, and buds.

Cultivar	All Phenolic Compounds		
	Leaves (mg/g ± SD)	Bark (mg/g ± SD)	Buds (mg/g ± SD)
‘Melrose’	98.93 ± 6.44	89.63 ± 1.45	105.82 ± 5.93
‘Melodie’	78.96 ± 11.81	81.18 ± 4.69	129.19 ± 11.10
‘James Grieve’	76.81 ± 3.65	99.86 ± 5.46	100.67 ± 7.21
‘Rubinola’	75.90 ± 3.04	89.26 ± 13.10	139.61 ± 4.52
‘Goldstar’	54.68 ± 10.38	83.62 ± 2.56	-
‘Meteor’	65.37 ± 7.92	70.81 ± 10.67	-
‘Průsvitné letní’	71.46 ± 2.80	78.75 ± 4.96	122.26 ± 1.19
‘Topaz’	106.81 ± 0.12	111.67 ± 3.48	133.01 ± 4.12
‘Red Bilt’	69.15 ± 4.05	101.35 ± 5.93	107.52 ± 4.77
‘Spartan’	103.07 ± 4.63	113.18 ± 2.60	-
‘Fragrance’	64.89 ± 1.27	100.64 ± 2.18	119.16 ± 6.76
‘Gloster’	104.72 ± 11.90	88.48 ± 2.32	101.05 ± 5.67
‘Bohemia Gold’	80.51 ± 6.26	84.69 ± 7.61	133.63 ± 11.47

Concentrations ± standard deviation (RSD, %) calculated from the mean of three measurements. ‘-’: The analysis was not performed.

**Table 3 pharmaceuticals-15-00244-t003:** Content of phenolic compounds in leaves of 13 cultivars (all values in mg/g of dried weight (DW)).

Cultivar	Phenolic Compound (mg/g ± SD)				
	Phloridzin	Phloretin	Chlorogenic Acid	Rutin	Quercitrin
‘Melrose’	91.17 ± 2.15	<LOQ	<LOQ	3.07 ± 0.56	3.27 ± 3.73
‘Melodie’	66.83 ± 2.82	<LOQ	0.44 ± 2.14	2.14 ± 3.98	9.37 ± 2.87
‘James Grieve’	72.14 ± 0.27	<LOD	0.97 ± 0.51	0.70 ± 2.53	2.91 ± 0.34
‘Rubinola’	60.10 ± 0.01	0.37 ± 0.11	0.32 ± 0.83	3.83 ± 1.69	11.29 ± 0.41
‘Goldstar’	46.43 ± 0.06	<LOQ	0.37 ± 7.84	1.96 ± 1.86	5.80 ± 0.61
‘Meteor’	61.69 ± 1.69	<LOQ	0.48 ± 4.81	1.16 ± 1.35	1.89 ± 0.14
‘Průsvitné letní’	62.19 ± 0.26	0.35 ± 0.35	5.58 ± 1.39	<LOQ	2.94 ± 0.81
‘Topaz’	94.93 ± 0.08	<LOQ	1.18 ± 0.44	1.36 ± 1.75	9.18 ± 2.51
‘Red Bilt’	60.74 ± 0.01	0.53 ± 0.01	0.52 ± 0.01	1.33 ± 0.06	6.03 ± 0.05
‘Spartan’	98.51 ± 0.22	0.28 ± 0.40	1.08 ± 1.61	1.10 ± 0.10	2.11 ± 2.30
‘Fragrance’	53.80 ± 0.05	<LOQ	<LOQ	3.39 ± 0.60	7.29 ± 0.62
‘Gloster’	93.80 ± 0.80	0.54 ± 0.54	0.76 ± 1.30	3.60 ± 8.26	6.02 ± 0.97
‘Bohemia Gold’	71.70 ± 0.26	<LOD	0.80 ± 0.37	1.77 ± 4.64	6.15 ± 0.98

Concentrations ± standard deviation (RSD, %) calculated from the mean of three measurements. nd: not detected. ‘-’: The analysis was not performed.

**Table 4 pharmaceuticals-15-00244-t004:** Content of phenolic compounds in bark of 13 cultivars (all values in mg/g of dried weight (DW)).

Cultivar	Phenolic Compound (mg/g ± SD)				
	Phloridzin	Phloretin	Chlorogenic Acid	Rutin	Quercitrin
‘Melrose’	76.59 ± 0.53	nd	nd	12.20 ± 0.64	0.84 ± 0.28
‘Melodie’	67.73 ± 0.09	nd	nd	11.41 ± 1.65	2.04 ± 2.95
‘James Grieve’	82.55 ± 0.59	nd	nd	16.03 ± 0.62	1.29 ± 4.26
‘Rubinola’	70.31 ± 0.70	nd	nd	16.54 ± 3.06	2.41 ± 9.35
‘Goldstar’	54.52 ± 0.40	nd	<LOQ	26.03 ± 1.29	2.85 ± 0.88
‘Meteor’	56.59 ± 1.70	nd	nd	13.49 ± 1.36	0.73 ± 7.60
‘Průsvitné letní’	57.48 ± 0.15	nd	nd	20.08 ± 0.29	1.19 ± 4.53
‘Topaz’	86.03 ± 0.41	nd	nd	22.58 ± 0.54	3.06 ± 2.53
‘Red Bilt’	85.62 ± 0.11	nd	nd	14.66 ± 0.60	1.07 ± 5.21
‘Spartan’	102.69 ± 0.55	nd	nd	12.29 ± 2.05	-
‘Fragrance’	82.50 ± 0.12	nd	nd	15.95 ± 70.71	2.19 ± 1.93
‘Gloster’	76.85 ± 0.17	nd	nd	10.68 ± 0.46	0.96 ± 1.69
‘Bohemia Gold’	65.08 ± 0.29	nd	nd	18.32 ± 2.38	1.28 ± 4.95

Concentrations ± standard deviation (RSD, %) calculated from the mean of three measurements. nd: not detected. ‘-’: The analysis was not performed.

**Table 5 pharmaceuticals-15-00244-t005:** Content of phenolic compounds in buds of nine cultivars (all values in mg/g of dried weight (DW)).

Cultivar	Phenolic Compound (mg/g ± SD)				
	Phloridzin	Phloretin	Chlorogenic Acid	Rutin	Quercitrin
‘Melrose’	94.41 ± 0.75	nd	3.48 ± 0.83	0.93 ± 2.39	7.00 ± 1.96
‘Melodie’	103.21 ± 1.34	nd	4.13 ± 1.33	3.22 ± 4.05	18.64 ± 4.39
‘James Grieve’	88.29 ± 1.27	nd	1.54 ± 1.21	2.41 ± 0.97	8.43 ± 3.76
‘Rubinola’	110.24 ± 0.67	nd	6.13 ± 0.73	2.80 ± 1.66	20.45 ± 1.46
‘Goldstar’	-	-	-	-	-
‘Meteor’	-	-	-	-	-
‘Průsvitné letní’	101.12 ± 0.55	nd	6.68 ± 0.51	1.93 ± 0.61	12.53 ± 0.23
‘Topaz’	111.68 ± 0.74	nd	1.93 ± 0.83	2.95 ± 2.00	16.45 ± 0.54
‘Red Bilt’	94.35 ± 1.20	nd	3.06 ± 1.32	1.80 ± 1.28	8.31 ± 0.96
‘Spartan’	-	-	-	-	-
‘Fragrance’	102.05 ± 1.43	nd	2.80 ± 1.43	1.12 ± 2.00	13.18 ± 1.90
‘Gloster’	88.51 ± 0.92	nd	2.69 ± 0.77	1.04 ± 2.16	8.82 ± 1.82
‘Bohemia Gold’	113.80 ± 2.85	nd	5.68 ± 3.17	3.07 ± 3.35	11.08 ± 2.10

Concentrations ± standard deviation (RSD, %) calculated from the mean of three measurements. nd: not detected. ‘-’: The analysis was not performed.

## Data Availability

The data presented in this study are available in article and Appendix A.

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
