# Peer review of "Determination of Phloridzin and Other Phenolic Compounds in Apple Tree Leaves, Bark, and Buds Using Liquid Chromatography with Multilayered Column Technology and Evaluation of the Total Antioxidant Activity"

_pharmaceuticals, 2022, doi:10.3390/ph15020244_

Round 1
Reviewer 1 Report
In this paper, the contents of five phenolic compounds in different parts of apple trees were determined and their antioxidant activities were comprehensively evaluated. Although the overall idea of this article is clear, the work is simple and many similar articles have been published, so I think that the article does not meet the criteria for publication in the Pharmaceuticals journal. The specific comments are as follows for the authors' reference.
- The title and keywords of the article do not reflect the part on evaluating antioxidant activity.
- It is recommended to add the purpose and significance of the study in the "Abstract".
- The order between the paragraphs in the "Introduction" is confusing, and it is recommended to revise it to fit the overall logic of the article.
- It is suggested to add the structural diagrams of these five phenolic compounds to the article.
- In lines 143-145, how do the authors explain that the reduction of the extracted plant mass and the increase of the extraction solvent resulted in a higher extraction yield and recovery? In general, the material-to-liquid ratio needs to be investigated systematically.
- In line 173, "5 um" should be changed to "5 μm".
- What is the meaning of the authors showing Figure 2 in the text? In section 2.4, the authors did not elucidate it clearly for the results. In addition, the ranges of the vertical coordinates of Figure 2 are inconsistent, which is difficult to put into the same dimension for comparison.
- In lines 195, 216, and 232, the authors keep using "Table 2" to indicate. In fact, it should be Table 3, 4, and 5, respectively.
- The phenolic profile in apple leaves, bark, and buds can be represented in a graph, which is more intuitive.
- It seems that critical conditions such as mobile phase and column temperature were not examined in this study.
- It is suggested that "Figure S10" in the supplementary material be placed in the main text.
- Have the raw plant materials of apple trees been authenticated by experts?
- It is recommended to emphasize the significance of this paper's research in the conclusion section.
- The author stated correction coefficients (r) in the table 1. Is this r or r-square (r2)?
- The resolution of the image can be improved appropriately.
Author Response
In this paper, the contents of five phenolic compounds in different parts of apple trees were determined and their antioxidant activities were comprehensively evaluated. Although the overall idea of this article is clear, the work is simple and many similar articles have been published, so I think that the article does not meet the criteria for publication in the Pharmaceuticals journal. The specific comments are as follows for the authors' reference.
We respect reviewer´s opinion, however, we (also reviewers 2 and 3) have a different one. One of the aim of this journal is “Biomolecules, natural products, phages, and cells as therapeutic tools”. The discovery of significant source of biomolecules and separation is the first step in the production of a drug as therapeutic tools, and their (routine) analysis is very important. Our article clearly provides information about the efficient method of extraction and separation of these biomolecules based on broad optimization steps.
- The title and keywords of the article do not reflect the part on evaluating antioxidant activity.
Based on this objective suggestion, the title and keywords are now corrected and include the term of antioxidant activity.
- It is recommended to add the purpose and significance of the study in the "Abstract".
The purpose of the study is to find a significance source of biomolecules and their effective separation. The main objective is to find substances that can be further utilize in (pharmaceutical, food) industry as is already written in the abstract “In this study, the various stationary phases for the determination of phenolic compounds in apple trees (leaves, bark, buds) were tested” and “This finding can lead to re-use of apple tree material to isolate substances that can be utilized in the food, pharmaceutical, or cosmetics industry.”. However, we added a few information.
- The order between the paragraphs in the "Introduction" is confusing, and it is recommended to revise it to fit the overall logic of the article.
The introduction section is structured in the following parts: Phloridzin – sources of phloridzin and other phenolic compounds – Properties of phloridzin and other phenolic compounds – HPLC method – column – electrochemical method – aim of study. On the other hand, we admit that some information was included in more parts. This is now corrected.
- It is suggested to add the structural diagrams of these five phenolic compounds to the article.
The structure of all tested compounds is now included in the manuscript.
- In lines 143-145, how do the authors explain that the reduction of the extracted plant mass and the increase of the extraction solvent resulted in a higher extraction yield and recovery? In general, the material-to-liquid ratio needs to be investigated systematically.
The final ratio plant material-liquid was investigated according to recovery parameter as is described in the manuscript. Lowering the weight of plant material resulted in better solubility of all compounds in the same volume of extraction solvent. Increasing the volume of extraction solvent did not affect the recovery. Moreover, the increase of extraction solvent volume resulted in more dilution of minor phenolic compounds and thus decreasing the signal for chlorogenic acid and phloretin. Therefore, the volume of 2 mL of solvent was finally used. This explanation was added in revised text.
- In line 173, "5 um" should be changed to "5 μm".
It was changed.
- What is the meaning of the authors showing Figure 2 in the text? In section 2.4, the authors did not elucidate it clearly for the results. In addition, the ranges of the vertical coordinates of Figure 2 are inconsistent, which is difficult to put into the same dimension for comparison.
The purpose of Figure 2 is informative - to show phenolic profile in real plant extract, while Fig. 1 shows the chromatography separation of standards only. The different ranges of y-axis at wavelengths of absorption maximum of analyzed compounds must be used due to the extremely different concentrations of phenolic compounds. E.g. concentration of phloridzin vs. chlorogenic acid is 100 times higher.
- In lines 195, 216, and 232, the authors keep using "Table 2" to indicate. In fact, it should be Table 3, 4, and 5, respectively.
In fact, mentioned lines really refer to Table 2, however, they simultaneously refer to Table 3, 4, and 5 for more detailed information about the content of individual phenolic compounds.
- The phenolic profile in apple leaves, bark, and buds can be represented in a graph, which is more intuitive.
This is a valid point. Therefore, we made 3 graphs and placed them to Supplementary material (Section 4) to avoid duplication in main text.
- It seems that critical conditions such as mobile phase and column temperature were not examined in this study.
According to our knowledge, mobile phase (MF) and column temperature (CT) is not critical for efficient resolution of phenolic compounds using reversed phase stationary phases. Nevertheless, we tested additionally different conditions of CT and starting ratio of organic phase for manuscript revision. The discussion of findings is added in revised text (Section 2.1., last part of the second column):
“During the short optimization, concentration of acetonitrile at the beginning of the gradient elution was tested in the range 2-20%. Low content of acetonitrile prolonged the whole analysis time while the higher content caused coelution of chlorogenic acid with peak of void volume on some stationary phases. Therefore, 10% of acetonitrile was found as optimal for screening of tested stationary phases. The column oven temperature was set to 30°C from the tested range 30-60°C. Higher temperature 60°C accelerated the last peak of phloretin from 10.45 min to 10.20 min. Thus, the shortening effect on whole analysis time was negligible. Therefore, higher temperature was not used because of the thermostability of silica-based columns.”
The separations under the initial conditions of gradient with 2% and 20% of organic phase are presented bellow:
- It is suggested that "Figure S10" in the supplementary material be placed in the main text.
Based on this suggestion, Figure S10 is now moved to manuscript.
- Have the raw plant materials of apple trees been authenticated by experts?
The raw plant materials were representatively collected by an academic staff. Authentication was confirmed by experts from RESEARCH AND BREEDING FRUIT INSTITUTE HOLOVOUSY s.r.o.
- It is recommended to emphasize the significance of this paper's research in the conclusion section.
The conclusion section was checked, and additional comments were revised.
- The author stated correction coefficients (r) in the table 1. Is this r or r-square (r2)?
It should be r2. It is now corrected.
- The resolution of the image can be improved appropriately.
We admit that the Figure 1 had poor quality. We therefore replaced it by new one with better resolution.
Reviewer 2 Report
The paper is interesting and, I hope, will be interesting for readers. The results are well documented, tables and figures are clear. Moreover, including part of results as supplementary material seems to be a good idea. Adequate number of references are well chosen,
However, the information regarding time of leaves and bark collection should be included into Material and Methods. Moreover, a dot is missing on line 39. Line 297 - value of g (centrifugation) should be added
Author Response
The paper is interesting and, I hope, will be interesting for readers. The results are well documented, tables and figures are clear. Moreover, including part of results as supplementary material seems to be a good idea. Adequate number of references are well chosen.
However, the information regarding time of leaves and bark collection should be included into Material and Methods. Moreover, a dot is missing on line 39. Line 297 - value of g (centrifugation) should be added.
The time of plant material collection is now included in Materials and Methods section: “…. were collected in location Hradec Králové, Roudnička, Czech Republic in July 2018 (leaves), September 2018 (bark), and February 2019 (buds).”
We added the missing dot and information about centrifugation in the manuscript.
Reviewer 3 Report
According to Turnitin, the manuscript appears to be original with a 26% similarity, and it may be of interest to journal readers. It is scientifically sound, and the data are valuable to the scientific community. However, authors should respond to specific questions for their manuscript to be reconsidered for publication. The following are my suggestions for improving the quality of the manuscript:
- The study established and validated a method for determining phloridzin, phloretin, chlorogenic acid, rutin, and quercitrin in apple tree leaves, bark, and buds using HPLC-DAD. The report detailed the validation procedure, which is pertinent to the title. However, references to analytical technique validation guidelines, such as AOAC, ICH, or others, should be included.
- Page 1, Line 17, L19; Page 4, Line 13: The SI system should be used to define units of measurement. ml or mL (no period) are both acceptable abbreviations. However, the authors should adhere to the same pattern throughout the paper.
- Page 2, Item 2.1. Optimization of the chromatographic separation conditions: the authors should provide the criteria to judge the symmetry of the peak from the tailing factor.
- Page 2, Line 23: The phrase "Chapter 1" is unclear. Please check and correct it.
- Page 3: In Figure 1, the peaks of the target compounds are asymmetry; please elaborate on the tailing factor interpretation.
- Page 4, L10: Add a period after the word "mixer".
- Pages 4-5, Table 1: Phloridzin has a linear range of 1,000–8,000 mg/L, whereas the other chemicals were examined in a concentration of 2–250 mg/mL. In this case, the LOD and LOQ levels of phloridzin were080 mg/L and 0.263 mg/mL, respectively. Kindly verify it in Table 1.
- Tables 2–5: The contents of the examined phenolic compounds in each apple cultivar are shown. Kindly include the significant differences between the groups for each compound.
- Page 4, Item 2.3. Validation of the method: The acceptable ranges for analytical performance parameters such as linearity, precision, accuracy, LOD, and LOQ as defined by worldwide guidelines such as AOAC, ICH, or others should be discussed.
- Page 8, Item 3.2 Raw plant material of apple trees: The authors should include the voucher specimen number and collection date for the analyzed plant.
- Page 9, Line numbers 4–5: The authors described that the concentration of the phloridzin standard was 500 mg/L and the remaining standards were 25 mg/L in the mixture. The preceding data contradict the linearity range (1,000–8,000 mg/L) for phloridzin stated in Table 1. Kindly check and clarify.
- Page 9, Item 3.5. HPLC equipment and analysis: The authors used the validated HPLC-DAD method to determine the phenolic component profile in various apple tree parts and cultivars. The Section Materials and Methods, on the other hand, is deficient in HPLC method validation. The authors should include a detailed description of the validation of the HPLC method.
- Page 9, line number 37: The phrase "Chapter 3.1" is unclear. Please check and correct it.
- Page 10, Item 4. Conclusions: Item 4. Conclusions: Please discuss the method's limitations and make recommendations for additional investigation.
- Table 1: "5 µm" instead of "5 um". ['µ' is a Greek letter]
- Page 9, Item 3.4. Extraction of phenolic compounds: The authors stated that 0.05 g of dried powdered materials were extracted in a 2 mL MeOH solution acidified with 0.1 percent (v/v) formic acid. However, the levels of phloridzin detected in Tables 2, 3, 4, and 5 were less than the linear range of this chemical's calibration curve. Please check and correct them all.
- Make 2 of R2 superscript in Table 1.
- An additional explanation should be given to establishing the distinction between this manuscript and the prior one (Dynamics of Phloridzin and Related Compounds in Four Cultivars of Apple Trees during the Vegetation Period).
Author Response
According to Turnitin, the manuscript appears to be original with a 26% similarity, and it may be of interest to journal readers. It is scientifically sound, and the data are valuable to the scientific community. However, authors should respond to specific questions for their manuscript to be reconsidered for publication. The following are my suggestions for improving the quality of the manuscript:
- The study established and validated a method for determining phloridzin, phloretin, chlorogenic acid, rutin, and quercitrin in apple tree leaves, bark, and buds using HPLC-DAD. The report detailed the validation procedure, which is pertinent to the title. However, references to analytical technique validation guidelines, such as AOAC, ICH, or others, should be included.
The validation parameters were made according to ICH guidelines that is now included in the manuscript.
- Page 1, Line 17, L19; Page 4, Line 13: The SI system should be used to define units of measurement. ml or mL (no period) are both acceptable abbreviations. However, the authors should adhere to the same pattern throughout the paper.
Based on this remark, we have unified the mL throughout the manuscript.
- Page 2, Item 2.1. Optimization of the chromatographic separation conditions: the authors should provide the criteria to judge the symmetry of the peak from the tailing factor.
The parameter of peak symmetry was calculated using validated chromatography software LAB Solutions, Shimadzu Co.. All values of chromatography system suitability test (SST) including peak symmetry are reported in Supplementary material for all columns (Table S2). Our validated method demonstrated the perfect peak symmetry for all compound ranging from 1.20 to 1.25. According the ICH validation rules, acceptable peak symmetry should be in range of 0.80 - 1.50. Therefore, our method fulfills these criteria.
- Page 2, Line 23: The phrase "Chapter 1" is unclear. Please check and correct it.
The term of chapter was replaced by a section.
- Page 3: In Figure 1, the peaks of the target compounds are asymmetry; please elaborate on the tailing factor interpretation
This is certainly a valid point. Poor image quality and worse visual resolution was caused during chromatogram copying from origin software. Now, it is corrected. As mentioned above and in Table S2, the peak symmetries were in the range of 1.20 - 1.25.
- Page 4, L10: Add a period after the word "mixer".
The period is added.
- Pages 4-5, Table 1: Phloridzin has a linear range of 1,000–8,000 mg/L, whereas the other chemicals were examined in a concentration of 2–250 mg/mL. In this case, the LOD and LOQ levels of phloridzin were080 mg/L and 0.263 mg/mL, respectively. Kindly verify it in Table 1
We verified the values of LOQ and LOD in Table 1 by dilution of new standards at concentration level of 0.5 mg/mL and their determination. The chromatographic patterns are enclosed bellow. All the quantified peaks are 10 times higher than signal noise to ratio. Thus, we experimentally confirmed the calculated values in Table 1.
The higher concentration range of phloridzin (1000 – 8 000 mg/L) was set according to its expected and relatively high occurrence in apple tree material as is mentioned in the manuscript.
- Tables 2–5: The contents of the examined phenolic compounds in each apple cultivar are shown. Kindly include the significant differences between the groups for each compound.
After discussion with statistician, we included the statistical evaluation of phloridzin concentration in leaves, bark, and buds. High values of phloridzin compared to other phenolic compounds are obvious (tens or hundreds of phloridzin vs. units of other phenolic compounds). The section of statistical evaluation is now commented in the revised text:
“There was found significant difference at level 0.05 in phloridzin concentrations among leaves, bark, and buds using the ANOVA test. By the Tukey method was found that concentration of phloridzin in buds is higher and different from leaves and bark, which have comparable concentrations.”
- Page 4, Item 2.3. Validation of the method: The acceptable ranges for analytical performance parameters such as linearity, precision, accuracy, LOD, and LOQ as defined by worldwide guidelines such as AOAC, ICH, or others should be discussed.
All the validation results fulfilled the criteria for quantification of phenolic compounds in real plant extracts. Calibration range was sufficient to determine wide range of different concentrations and LOQ of the method was sufficiently sensitive to calculate real low concentrations. Other parameters fulfill the standard validation criteria for analysis of raw plant samples. The sentence was added to revised text.
- Page 8, Item 3.2 Raw plant material of apple trees: The authors should include the voucher specimen number and collection date for the analyzed plant.
Raw plant material from apple trees was collected from the location of Hradec Králové, Roudnička, which is mentioned in the manuscript. The collection date is now added to the manuscript. However, the analyzed apple trees do not contain specimen number. Nevertheless, authentication was confirmed by experts from RESEARCH AND BREEDING FRUIT INSTITUTE HOLOVOUSY s.r.o
- Page 9, Line numbers 4–5: The authors described that the concentration of the phloridzin standard was 500 mg/L and the remaining standards were 25 mg/L in the mixture. The preceding data contradict the linearity range (1,000–8,000 mg/L) for phloridzin stated in Table 1. Kindly check and clarify.
The concentration of phloridzin 500 mg/L was used for method development and optimization. The calibration range 1,000 – 8,000 mg/L was used for validation of the method due to the extremely high content of phloridzin in extracts. We admit it could be confusing. Therefore, we clarified it in the manuscript, where the section 3.3 is - preparation of the standard solutions for method optimization, and newly added section 3.6 is - Validation of the method.
- Page 9, Item 3.5. HPLC equipment and analysis: The authors used the validated HPLC-DAD method to determine the phenolic component profile in various apple tree parts and cultivars. The Section Materials and Methods, on the other hand, is deficient in HPLC method validation. The authors should include a detailed description of the validation of the HPLC method.
The Method validation section (3.6) is now added to Materials and Methods section.
- Page 9, line number 37: The phrase "Chapter 3.1" is unclear. Please check and correct it.
It is corrected now.
- Page 10, Item 4. Conclusions: Item 4. Conclusions: Please discuss the method's limitations and make recommendations for additional investigation.
The Conclusion section was revised.
- Table 1: "5 µm" instead of "5 um". ['µ' is a Greek letter].
It was changed.
- Page 9, Item 3.4. Extraction of phenolic compounds: The authors stated that 0.05 g of dried powdered materials were extracted in a 2 mL MeOH solution acidified with 0.1 percent (v/v) formic acid. However, the levels of phloridzin detected in Tables 2, 3, 4, and 5 were less than the linear range of this chemical's calibration curve. Please check and correct them all.
Tables 2 – 5 contain final amounts of phloridzin converted to 1 gram of dry material (mg/g DW) (the concentration of a determined compound was converted to the amount of used extraction solvent followed by the conversion to the real sample weight with the correction to the purity of the used standard). Calibration range of phloridzin used for validation was 1000 – 8000 mg/L because the concentrations of phloridzin in final extracts were extremely high. According to our knowledge, these are two different things and no correction is needed.
- Make 2 of R2 superscript in Table 1.
It is already made.
- An additional explanation should be given to establishing the distinction between this manuscript and the prior one (Dynamics of Phloridzin and Related Compounds in Four Cultivars of Apple Trees during the Vegetation Period).
The study of Táborský et al. was intended to observe phloridzin during vegetation period. Only 4 phenolic compounds were observed in 4 cultivars. We determine 5 compounds in 13 cultivars. Its aim was to find a period with the highest values of phloridzin. Analysis of the phenolic compounds profile and screening of various cultivars was not the objective of the research. This study did not focused on method optimization, validation, or extraction process like us. Moreover, the determination of antioxidant activity and the comparison with HPLC was not performed at all.
Round 2
Reviewer 1 Report
There are several other questions about this manuscript, as follows.
- The title of the article is too wordy, and the specific compounds can be summarized as phenolic compounds.
- Even if it is an unstructured abstract, its content should include four parts: purpose, methods, results, and conclusions. In lines 26-27, total antioxidant activity was correlated with the total amount of phenolic compounds, not HPLC. The main results regarding its antioxidant activity were not presented in the abstract. Lines 24-25 belong to the conclusion section and are best placed at the end.
- The chemical structures in Figure 1 should be re-drawn with ChemDraw software, rather than intercepting diagrams from other literatures.
- In section 3.2, the fact that apple trees have been identified by experts is best mentioned in the text.
Author Response
Reviewer #1:
Comments and Suggestions for Authors:
There are several other questions about this manuscript, as follows.
- The title of the article is too wordy, and the specific compounds can be summarized as phenolic compounds.
Based on this objective suggestion, the title is now shortened. We replaced “Determination of phloridzin, phloretin, chlorogenic acid, rutin, and quercitrin in apple tree leaves, bark, and buds using liquid chromatography with multilayered column technology and evaluation of the total antioxidant activity” by “Determination of phloridzin and other phenolic compounds in apple tree leaves, bark, and buds using liquid chromatography with multilayered column technology and evaluation of the total antioxidant activity”.
We would like to keep the term of phloridzin because it is the major compound of tested material and is discussed in the manuscript.
- Even if it is an unstructured abstract, its content should include four parts: purpose, methods, results, and conclusions. In lines 26-27, total antioxidant activity was correlated with the total amount of phenolic compounds, not HPLC. The main results regarding its antioxidant activity were not presented in the abstract. Lines 24-25 belong to the conclusion section and are best placed at the end.
We corrected the abstract that is now slighty restructured according to reviewer´s suggestion and is more precisely divided in the following parts: purpose “Apples are known to be a rich source of phenolic compounds, however detailed studies about their content in individual parts of apple trees are reported rarely. For this purpose, we tested various stationary phases for the determination of phenolic compounds in leaves, bark, and buds”, method “Phloridzin, phloretin, chlorogenic acid, rutin, and quercitrin - were analyzed with high performance liquid chromatography coupled with diode array detection. A YMC Triart C18-ExRS 150 × 4.6 mm, 5 µm particle size analytical column with multilayered particle technology was used. The separation was performed with a mobile phase consisted of acetonitrile and 0.1% phosphoric acid according to the gradient program at a flow rate of 1 mL/min in 12.50 minutes”, results “The concentration of phenolic compounds from 13 cultivars was in the range of 64.89 – 106.01 mg/g of dry weight (DW) in leaves, 70.81 – 113.18 mg/g DW in bark, and 100.68 – 139.61 mg/g DW in buds. Phloridzin was a major compound. Total antioxidant activity was measured using flow analysis and the correlation with the total amount of phenolic compounds was found”, and conclusion “This finding can lead to re-use of apple tree material to isolate substances that can be utilized in the food, pharmaceutical, or cosmetics industry”.
The basic information (result) regarding antioxidant activity is the correlation with the total amount of phenolic substances that is mentioned in the abstract. Since this is not the main subject of our study, we think that this information is sufficient for the abstract.
- The chemical structures in Figure 1 should be re-drawn with ChemDraw software, rather than intercepting diagrams from other literatures.
Based on this suggestion we replaced the original structures by new ones from the standard Sigma-Aldrich web database page to avoid interpreting pictures from other literatures.
- In section 3.2, the fact that apple trees have been identified by experts is best mentioned in the text.
The section 3.2. now contains “Authentication was confirmed by experts from Research and Breeding Institute of Pomology Holovousy Ltd.“
In addition, the introduction section was restructured.
Reviewer 3 Report
The reviewer thanks the authors for the review work done and the attention given to the remarks made. The work carried out has correctly taken into account the requests made.
Author Response
Reviewer #3:
Comments and Suggestions for Authors:
The reviewer thanks the authors for the review work done and the attention given to the remarks made. The work carried out has correctly taken into account the requests made.
Thank you. We have no other comments.
Round 3
Reviewer 1 Report
The manuscript is improved after addressing all the comments.
Author Response
Thank you. We have no other comments.